



# Atmospheric breakdown kinetics and air quality impact of potential "green" solvents the oxymethylene ethers OME3 and OME4

James D. D'Souza Metcalf[1,2], Ruth K. Winkless[1], Caterina Mapelli[1,2,3], C. Rob McElroy[2,4], Claudiu Roman[5,6], Cecilia Arsene[5,6,7], Romeo I. Olariu[5,6,7], Iustinian G. Bejan[5,7] and Terry J. Dillon[1].

[1]Wolfson Atmospheric Chemistry Laboratories, Department of Chemistry, University of York, York, YO10 5DD, UK.
[2]Green Chemistry Centre of Excellence, Department of Chemistry, University of York, York, YO10 5DD, UK
[3]now at: National Reasearch Council - Institute of Methodologies for Environmental Analysis (IMAA), Tito Scalo, Potenza, 85050, Italy
[4]now at: Department of Chemistry, School of Natural Sciences, University of Lincoln, Brayford Pool, Lincoln, LN6 7TS, UK
[5]Faculty of Chemistry, "Alexandru Ioan Cuza" University of Iasi, 700506, Iasi, Romania
[6]Research Center with Integrated Techniques for Atmospheric Aerosol Investigation in Romania (RECENT-AIR), "Alexandru Ioan Cuza" University of Iasi, 11th Carol I, Iasi, 700506, Romania
[7]Integrated Center of Environmental Science Studies in the North Eastern Region – CERNESIM, "Alexandru Ioan Cuza" University of Iasi, 700506, Iasi, Romania

*Correspondence to*: Terry J. Dillon (terry.dillon@york.ac.uk) and Iustinian G. Bejan (iustinian.bejan@uaic.ro)

**Abstract.** Laboratory-based experiments were used to investigate the atmospheric degradation chemistry of two oxymethylene ethers, $CH_3O(CH_2O)_3CH_3$ (OME3) and $CH_3O(CH_2O)_4CH_3$ (OME4). OME3 and OME4 have been proposed as promising "green" replacement compounds for problematic ethereal solvents such as 1,4-dioxane and tetrahydrofuran. Results from direct, absolute laser-based experiments and from a series of complementary relative rate studies demonstrated that OH + OME3 (R3) proceeded with a rate coefficient $k_3(296 \text{ K}) = (1.0 \pm 0.2) \times 10^{-11} \text{ cm}^3 \text{ molecule}^{-1} \text{ s}^{-1}$, a factor of two smaller than predicted by structure activity relationships (SAR). Evidence for a complex mechanism was provided by $k_3(294 - 464 \text{ K})$, characterised by deviations from Arrhenius-like behaviour close to room temperature. A further series of relative rate experiments were used to determine a rate coefficient of $k_4(296 \text{ K}) = (1.1 \pm 0.4) \times 10^{-11} \text{ cm}^3 \text{ molecule}^{-1} \text{ s}^{-1}$ for OH + OME4. These results allowed for lifetimes, $\tau \approx 1$ day to be estimated for the removal of each of OME3 and OME4 from the troposphere. Photochemical Ozone Creation Potential estimates ($POCP_E$) were calculated for NW-Europe conditions. These were considerably smaller than equivalent metrics for the problematic solvents they may replace, largely owing to their lack of C-C bonds. In the course of this work, rate coefficients (in $10^{-11} \text{ cm}^3 \text{ molecule}^{-1} \text{ s}^{-1}$) were determined for Cl + OME3, $k_6(296 \text{ K}) = (17 \pm 4)$ and for Cl + OME4, $k_7(296 \text{ K}) = (19 \pm 6)$.

## 1   Introduction

As traditional sources of volatile organic compound (VOC) emissions are subject to increasingly strict regulation, new and formerly overlooked emission routes are growing in significance. Of these, solvents are emerging as the dominant anthropogenic source of non-methane VOC (Lewis et al., 2020). Simultaneously, the landscape of solvent use and emission



is undergoing a shift as research and industry move away from harmful and environmentally damaging petroleum derived

solvents and in their place safer, renewable bio-based alternatives are being developed as part of the move towards net zero (Ashcroft et al., 2015; Bryan et al., 2018; Constable et al., 2007). While fundamental air quality metrics are discussed in some solvent selection guides and development protocols, there remains a significant and troubling lack of knowledge surrounding the atmospheric fate of newly developed "green" solvents.

The oxidative breakdown of VOC in air is known to yield harmful ozone ($O_3$), formaldehyde (HCHO), and particulates

(Bloss et al., 2005; Hansen et al., 1975; Saunders et al., 2003). Waste and inefficient use of volatile solvents is therefore a well-established source of harmful emissions to the atmosphere. Poor air quality  has been estimated to cause of over 400,000 annual deaths in Europe alone (European Environment Agency., 2020; Lelieveld et al., 2019). Ethers form a significant portion of atmospheric non-methane VOCs, and are emitted almost entirely from anthropogenic sources (Calvert et al., 2015). Whilst unsaturated ethers are known to be significantly more reactive than unsaturated ethers, the latter are far

more widely used in solvent applications (Zhou et al., 2006). Many ethereal solvents, examples of which include 1,4-dioxane (1,4-dioxacyclohexane, $C_4H_8O_2$, henceforth dioxane) and tetrahydrofuran (oxolane, $C_4H_8O$, henceforth THF) are manufactured from unsustainable petrochemical feedstocks, harmful to health (potentially carcinogenic), form dangerous peroxides and are environmentally hazardous (Prat et al., 2016). Despite this, analysis of trends in solvent usage have  shown that dioxane and THF consistently remain among the most widely used solvents in newly reported processes (Ashcroft et al.,

2015; Jordan et al., 2021). Considerable research effort has therefore been directed towards developing sustainable and safe new "green" solvents (de Gonzalo et al., 2019; Jordan et al., 2022). Oxymethylene ethers ($CH_3O(CH_2O)nCH_3$, henceforth OME, specific ethers an OME$n$) can be synthesised at scale from readily available, renewable and bio-derivable methanol ($CH_3OH$) and formaldehyde (HCHO) (Peter et al., 2018). With carbon capture and utilisation, circular low-carbon $CO_2$ derived "e-methanol" is also a potential commercially available large-scale feedstock. Formaldehyde from $CO_2$ is also being

extensively researched  (Zhao et al., 2022). Numerous studies have shown that OME blends are viable as carbon neutral diesel alternatives, associated with low particulate generation due to their lack of carbon-carbon bonds (Fenard and Vanhove, 2021; Härtl et al., 2017; Jacob and Maus, 2017; Schmitz et al., 2022; Sun et al., 2025; Zhang et al., 2014). OME have also been demonstrated as safer, bio-based alternatives to traditional ethereal solvents (Zhenova et al., 2019).

A crucial, often rate-determining step in atmospheric VOC oxidation mechanisms is the initial breaking of a C-C or C-H

bond, either directly *via* photolysis or following attack by oxidants such as $O_3$ or gas-phase free-radicals (Seinfeld, 2016). Dioxane, THF and the OME are all saturated VOC and lack a near-UV chromophore. As such their principal breakdown route is likely *via* bimolecular reaction with the principal gas-phase oxidant, the hydroxyl radical (OH), e.g. (R1 – R4):

$$OH + dioxane \rightarrow (products) \qquad (R1)$$

$$OH + THF \rightarrow (products) \qquad (R2)$$

$$OH + OME3 \rightarrow (products) \qquad (R3)$$

$$OH + OME4 \rightarrow (products) \qquad (R4)$$



Rate coefficients, $k$(298 K) for OH-mediated removal of traditional volatile solvents are reasonably well established. The available literature for $k_1$ (Dagaut et al., 1990; Maurer et al., 1999; Porter et al., 1997) was most recently supplemented by Moriarty et al (2003) who estimated a lifetime $\tau_1 \approx 25$ hours for removal of dioxane from the troposphere. Literature for $k_2$

(Illés et al., 2021; Ravishankara and Davis, 1978; Wallington et al., 1988; Winer et al., 1977) is in agreement on a more rapid removal process for THF, for which Moriarty et al estimated $\tau_2 \approx 16$ hours. By contrast with these more established solvents, there appears to be no gas-phase data on (R3 – R4), from which to estimate lifetimes and other air quality metrics. In the absence of such data, even a cursory assessment of the environmental impact of replacing dioxane or THF with OME is not feasible. In the absence of experimental data, $k$-values may be predicted by Structure Activity Relationships (SAR)

such as those most recently formulated by Jenkin et al. (2018). However, the lack of literature data available for OH + ethers results in SAR predictions with a large associated uncertainty. For example, a recent experimental study by demonstrated that the atmospheric breakdown of the new "green" solvent 2,2,5,5-tetramethyloxolane proceeds a factor of three slower than SAR prediction (Mapelli et al., 2022).

$$\text{OH} + \text{TMO} \rightarrow \text{(products)} \tag{R5}$$

Accordingly, the objectives of this work were to use lab-based experiments to determine accurate values of $k3$ and $k4$ and to interpret these results in terms of atmospheric lifetimes and impacts on air quality. In the course of this work, rate coefficients were determined for reactions (R6-R7) of OME with another atmospheric oxidant, atomic chlorine. These provide a further indication into the atmospheric fate of OMEs in regions where chlorine is significant (Ariya, 1999; Atkinson and Aschmann, 1985; Thornton et al., 2010).

$$\text{Cl} + \text{OME3} \rightarrow \text{(products)} \tag{R6}$$

$$\text{Cl} + \text{OME4} \rightarrow \text{(products)} \tag{R7}$$

## 2  Experimental

Laboratory-based experiments were carried out using two well-established kinetic techniques. Following a description of the isolation and characterisation of OME samples in section 2.1, section 2.2 describes the ESC-Q-UAIC environmental

simulation chamber at Alexandru Ioan Cuza University of Iasi, Romania, used for relative rate determinations of $k_3$(296 K) and $k_4$(296 K). The pulsed laser photolysis (PLP) apparatus used for direct, absolute determinations of $k_3$(294 – 464 K) at University of York, UK, is described in section 2.3. Computational procedures used to estimate structural, spectral and kinetic parameters are outlined in section 2.4.

### 2.1  OME Separation and Characterisation

OME3 (2,4,6,8-tetraoxanonane) and OME4 (2,4,6,8,10-Pentaoxaundecane) were isolated from an OME blended fuel mix (ChemCom Industries) by vacuum distillation. Fractions were identified by NMR and ESI-MS analysis, with sample purities



of > 97% estimated *via* GC-FID. OME5 was similarly isolated and identified, but its vapour pressure was too low to be of utility in this work. Spectra and chromatograms of isolated fractions are contained in the supplementary information (S3).

## 2.2 Smog Chamber Studies

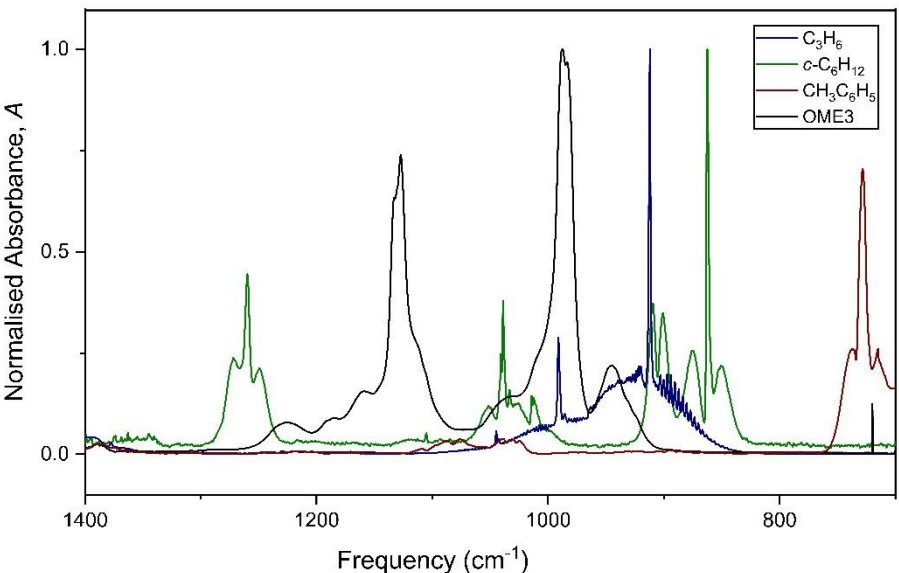


**Figure 1: Spectra of OME3 and reference compounds taken in ESC-Q-UAIC chamber. Compounds were monitored at, 912 cm$^{-1}$ (C$_3$H$_6$), 862 cm$^{-1}$ (*c*-C$_6$H$_{12}$), 729 cm$^{-1}$ (CH$_3$C$_6$H$_5$) and 989 cm$^{-1}$.**

Relative rate experiments were conducted in a 760 dm$^3$ quartz chamber, equipped with inlet ports, sampling lines, two sets of UV lamps (254 and 365 nm), and multi-pass FTIR instrumentation for monitoring of precursors, OME, reference VOC and
oxidation products as described in recent publications (Roman et al., 2022). Liquid samples were supplied to the reactor by direct injection through a septum: H$_2$O$_2$ (35% in H$_2$O, Sigma-Aldrich); OME3 and OME4 (see section 2.1); CH$_3$C$_6$H$_5$ 99.5% (Carl Roth), *c*-C$_6$H$_{12}$ 99.5% (Sigma-Aldrich), C$_3$H$_6$ >99.9% (Sigma-Aldrich) and C$_2$O$_2$Cl$_2$ 98% (Sigma-Aldrich) were used as supplied. All experiments were conducted at $p$ = (1000 ± 10) mbar (air) and $T$ = (296 ± 2) K. OH was generated *via* the direct 254 nm photolysis of H$_2$O$_2$ (R8).

$$H_2O_2 + h\nu \rightarrow 2OH \quad\quad\quad\quad\quad (R8)$$

For all $k_{6-7}$ determinations, atomic chlorine was generated by the photolysis of oxalyl chloride at 254 nm (R9) (Stuhr et al., 2024).

$$(COCl)_2 + h\nu \rightarrow\rightarrow 2Cl + 2CO \quad\quad\quad\quad\quad (R9)$$

Experiments conducted in the absence of radical precursors demonstrated that neither OME3 nor the various reference
compounds were significantly impacted by wall losses or photolytic removal, indicating that only small corrections to





subsequent kinetic data were necessary. Corrections were required (up to 20%) when processing OME4 data. These corrections were applied in the manner we have previously reported (Mairean et al., 2024). FTIR peak intensities were directly proportional to species concentrations and were used to calculate the logarithmic depletion for OME and for the reference compound. According to Eq. (1), the slope obtained from a proportional fit may be identified with the relative rate

$k_3$ / $k_{ref}$.

$$ln\left(\frac{[OMEn]_{t0}}{[OMEn]_t}\right) - \left(k_{photo} + k_{wall}\right)(t - t_0) = \left(\frac{k_3}{k_{ref}}\right)ln\left(\frac{[reference]_{t0}}{[reference]_t}\right) \quad (1)$$

Where $k_{photo}$ and $k_{wall}$ are photolysis frequency ($J$) and rate of wall loss respectively, measured under experimental conditions for OME4 and assumed to be zero for all other compounds.

Reference compounds were selected to satisfy two main criteria. Firstly, it was desirable for the FITR spectra of references
to have well defined peaks that were close to, but overlapped minimally with, those of the OME (Fig. 1). This was achieved by comparing reference spectra from the literature to OME spectra predicted *via* DFT calculations (section 2.4). Secondly, reference compound $k$(296 K) data needed to be well-established in the literature (see Table 1); these values were preferably of a similar magnitude to measured or predicted target $k$-values. According to these principles the selected reference compounds were toluene ($CH_3C_6H_5$), propene ($C_3H_6$) and cyclohexane ($c$-$C_6H_{12}$), as represented in Table 1.

**Table 1: Eurochamp recommended kinetic data (McGillen et al., 2020) for reference compounds used in rate coefficient determination in this work.**

| Reaction | Reference Reaction | $k_{298}$ / $10^{-12}$ cm$^3$ molecule$^{-1}$ s$^{-1}$ | Source of recommended values |
|:---:|:---:|:---:|:---:|
| (R10) | $C_3H_6$ + OH | 24.4 ± 3.66 | Fitted or manually entered data from multiple sources (McGillen et al., 2020) |
| (R11) | $c$-$C_6H_{12}$ + OH | 6.69 ± 0.67 | Fitted or manually entered data from multiple sources (McGillen et al., 2020) |
| (R12) | $CH_3C_6H_5$ + OH | 5.60 ± 1.46 | Evaluated kinetic and photochemical data for atmospheric chemistry (IUPAC) (Mellouki et al., 2021) |
| (R13) | $C_3H_6$ + Cl | 270 ± 30.0 | Evaluated kinetic and photochemical data for atmospheric chemistry (IUPAC) (Atkinson et al., 2006) |
| (R14) | $c$-$C_6H_{12}$ + Cl | 330 ± 49.5 | (Calvert et al., 2015) |
| (R15) | $CH_3C_6H_5$ + Cl | 60.0 ± 6.00 | (Calvert et al., 2015) |

Rate coefficients were taken from the EUROCHAMP Database for the Kinetics of the Gas-Phase Atmospheric Reactions of Organic Compounds version 3.1.0. (McGillen et al., 2020). FTIR spectra were recorded every minute by combining 114
scans for a spectrum, with approximately 30 such spectra at a resolution of 1 cm$^{-1}$ completing each experiment.



## 2.3 PLP-LIF Experiments

Absolute determinations of $k_3$(294 – 502 K) were carried out in York using the PLP apparatus, detailed in recent publications (Mapelli et al., 2022, 2023). Briefly, output from a Nd:YAG laser (Quantel, Q-Smart, 20 mJ / 266 nm pulse at 10 Hz) was directed into a 400 cm3 pyrex reactor and used to generate OH *via* $H_2O_2$ photolysis (R8). Output from a frequency-doubled

dye laser (Radiant dyes, Rhodamine-6G) was used to pump the Q11 transition of OH at 281.997 nm. Off-resonant 308 nm fluorescence was collected by a photomultiplier (Hamamatsu) to effect LIF detection of OH. Reactor temperature was regulated by heating tape and monitored by a thermocouple. Four calibrated mass flow controllers regulated gas flow rates with pressure monitored by a calibrated capacitance manometer. Such absolute rate coefficient determinations relied upon accurate knowledge of [OME]. In the case of OME3, mixtures were prepared on a vacuum line by mixing solvent vapours ≈

0.5% in 1300 mbar $N_2$ for storage in 12 dm$^3$ pyrex bulbs. [OME3] values were calculated to ± 15% estimated precision from manometric measurements. Data conducted at [OME3] = 0, together with manometric estimates indicated that [H2O2] < $10^{14}$ molecule cm$^{-3}$ and consequently that [OH] ≈ 5 × $10^{11}$ molecule cm$^{-3}$ was generated (R8) in each experiment. Pseudo first-order conditions, [OME3] >> [OH] and / or [H2O2] >> [OH] therefore applied throughout.

Handling of OME4 was more of a challenge. A vapour pressure of $p_{vap}$ = (0.62 ± 0.15) mbar was measured in a series of

experiments conducted on the vacuum line at $T$ = 296 K. OME4 volatility was consequently too low for reliable preparation and delivery *via* pyrex bulbs. OME4 was therefore supplied *via* flow of $N_2$ through a pyrex bubbler located upstream of the reactor. Modest flow rates of 0 - 30 sccm ($N_2$) were used to best aid saturation of [OME4] as the bubbler conditions were maintained at p ≈ 100 mbar and (with use of a thermostatic water bath) $T$ = 296 K.

Chemicals: $N_2$ > 99.9999%, obtained from liquid nitrogen boil-off and $O_2$ (99.995%, BOC) were used as supplied; $H_2O_2$ (JT

Baker, 60% in $H_2O$) was pre-concentrated by continuous flow of $N_2$ to remove $H_2O$ then supplied *via* a bubbler; OME3 was isolated from other OME fractions (see section 2.1) then subjected to repeat T = 77 K freeze-pump-thaw cycles to remove air prior to dilution in $N_2$ and storage.

## 2.4 Calculations

Gaussian Calculations: IR spectrum prediction was performed using Gaussian 16 (Frisch et al., 2016), from an input created

with Gaussview (version 6.1.1, Semichem). For each molecule an optimisation and frequency calculation was performed at the B3LYP level using a 6-31G basis set. IR spectra were generated in Gaussview from the predicted vibrational energies of the molecule in the Gaussian output file. Optimised structures and input and output files are available in the supplementary information (S1).

COSMO-RS calculations: Chem3D (version 21.0.0, PerkinElmer) was used to generate approximate coordinates of

compounds. Energy minimisation on these structures was carried out using the inbuilt MM2 calculation tool to a minimum RMS gradient of 0.010. These minimised structures were then transferred to COSMOconfX (Version 4.0, COSMOlogic GmbH & Co. KG) as .mol files. Confirmation and energy calculations were then carried out using the BP-TZVP-GAS





template. The .cosmo and .energy files were transferred to COSMOthermX (Eckert and Klamt, 2016) and rate coefficients were estimated using a local MOPAC7 implementation (MOOH) derived from that previously described by Klamt (Klamt,

1993, 1996), *via* the environmental properties menu.  Optimised structures and input and output files are available in the supplementary information (S1.2).

## 3   Results and Discussion

Results from two experimental studies are described below. In section 3.1, the results from ambient temperature relative-rate experiments are presented. Details of complementary determinations of $k_3$(294 – 464 K) in direct, absolute laser-based

experiments are presented in section 3.2. Section 3.3 describes results from relative rate determinations of $k_5$ and $k_6$ (reactions of OME with Cl-atoms). The discussion section 3.4 attempts to rationalise these results in the context of predictions from structure activity relationships and the limited literature dataset for similar oxygenated VOC. Uncertainties (±) quoted throughout this work are two sigma statistical only, derived from regression analysis, unless specifically stated otherwise. In the case of relative determinations quoted uncertainties also account for the reported uncertainty in the

reference rate constant, by the method we have previously described (Mairean et al., 2024).

### 3.1   Relative Rate Determination of $k$(296 K)

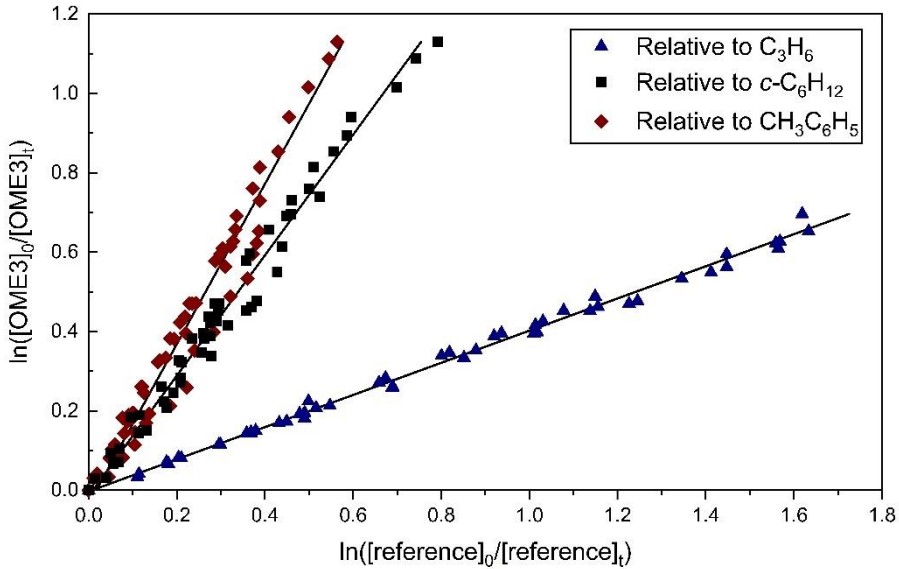

**Figure 2: Exemplar plot, used to determine $k_{3RR}$(296 K) for OH + OME3. Blue triangles correspond to use of $C_3H_6$ (R10) as reference; the black squares $c$-$C_6H_{12}$ (R11) and the red diamonds $CH_3C_6H_5$ (R12). The solid lines are linear fits, the gradients of**
**which were used to obtain $k_3$ (Eq.3) in conjunction with literature data (Table 1). Each experiment was repeated twice, and the resulting values of $k_3$ (Table 2) were averaged to obtain $k_{3,RR}$(296 K) = (1.04 ± 0.09) × 10$^{-11}$ cm$^3$ molecule$^{-1}$ s$^{-1}$.**





Figure 2 displays the results of studies using propene ($C_3H_6$), cyclohexane ($c$-$C_6H_{12}$) and toluene ($CH_3C_6H_5$) as reference compounds to determine $k_3$(296 K). These data demonstrate good linearity across a wide range of relative reactant concentrations. Two experiments were carried out per reference compound for both OME3 and OME4, the results of which are presented in Table 2. The spread of values appears reasonable given both the systematic uncertainties in the various literature reference rate coeffficients and statistical uncertainites. Weighted mean values from all relative rate determinations were $k_{3,RR}$(296 K) $= (1.04 \pm 0.09) \times 10^{-11}$ cm$^3$ molecule$^{-1}$ s$^{-1}$ and $k_{4,RR}$(296 K) $= (1.11 \pm 0.10) \times 10^{-11}$ cm$^3$ molecule$^{-1}$ s$^{-1}$.

**Table 2: Results of relative determinations of $k_3$(296 K), $k_4$(296 K), $k_6$(296 K) and $k_7$(296 K) for each reference compound.**

| Reaction | Reference | $k / k_{ref}$ | $k$(296 K) $10^{-11}$ cm$^3$ molecule$^{-1}$ s$^{-1}$ |
|:---:|:---:|:---:|:---:|
| R3 | $C_3H_6$ + OH (R10) | $0.40 \pm 0.005$ | $0.98 \pm 0.15$ |
| R3 | $c$-$C_6H_{12}$ + OH (R11) | $1.61 \pm 0.037$ | $1.08 \pm 0.11$ |
| R3 | $CH_3C_6H_5$ + OH (R12) | $1.96 \pm 0.032$ | $1.10 \pm 0.29$ |
| R4 | $C_3H_6$ + OH (R10) | $0.42 \pm 0.015$ | $1.02 \pm 0.16$ |
| R4 | $c$-$C_6H_{12}$ + OH (R11) | $1.71 \pm 0.040$ | $1.14 \pm 0.12$ |
| R4 | $CH_3C_6H_5$ + OH (R12) | $2.08 \pm 0.036$ | $1.16 \pm 0.30$ |
| R6 | $C_3H_6$ + Cl (R13) | $0.63 \pm 0.021$ | $17.1 \pm 2.2$ |
| R6 | $c$-$C_6H_{12}$ + Cl (R14) | $0.49 \pm 0.014$ | $16.1 \pm 2.6$ |
| R6 | $CH_3C_6H_5$ + Cl (R15) | $2.81 \pm 0.043$ | $16.7 \pm 1.8$ |
| R7 | $C_3H_6$ + Cl (R13) | $0.70 \pm 0.014$ | $18.9 \pm 2.2$ |
| R7 | $c$-$C_6H_{12}$ + Cl (R14) | $0.55 \pm 0.012$ | $18.0 \pm 2.8$ |

Determinations of $k_6$ and $k_7$ at $T$ = 296 K were carried out in a similar manner, with the exception of only two references being used to determine $k_7$ ($C_3H_6$ and $c$-$C_6H_{12}$) due to difficulties with overlapping peaks in $CH_3C_6H_5$ experiments. An exemplar relative rate plot for $k_{6,RR}$(296 K) is shown in Figure 3. Weighted mean values from all such relative rate determinations were $k_{6,RR}$(296 K) $= (1.68 \pm 0.12) \times 10^{-10}$ cm$^3$ molecule$^{-1}$ s$^{-1}$ and $k_{7,RR}$(296 K) $= (1.85 \pm 0.18) \times 10^{-10}$ cm$^3$ molecule$^{-1}$ s$^{-1}$. Full experimental data including plots and fits are available in the supplementary information (S4).



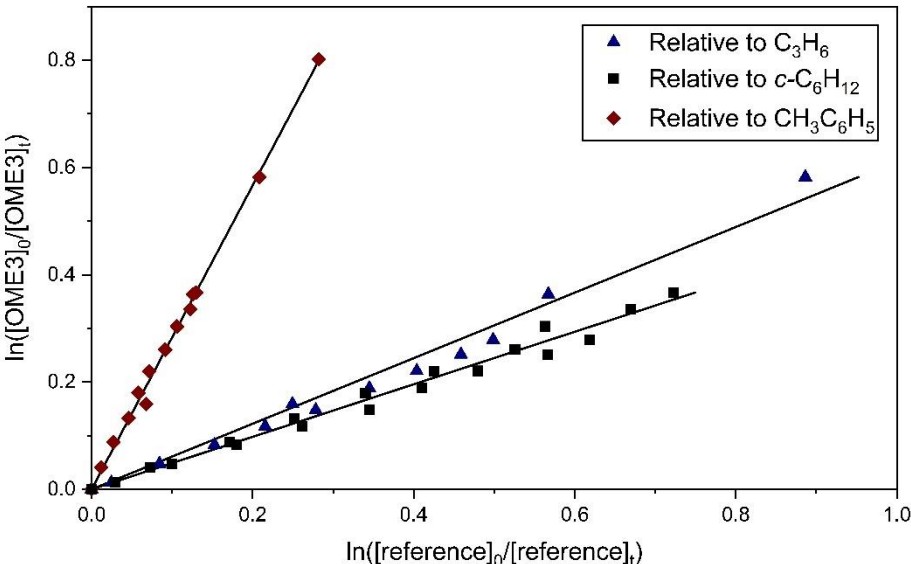

**Figure 3: Exemplar plot, used to determine $k_{6RR}$(296 K) for Cl + OME3. Blue triangles correspond to use of $C_3H_6$ (R10) as reference; the black squares $c$-$C_6H_{12}$ (R11) and the red diamonds $CH_3C_6H_5$ (R12). The solid lines are linear fits, the gradients of which were used to obtain $k_6$ (Eq.3) in conjunction with literature data (Table 1). Each experiment was repeated twice, and the resulting values of $k_6$ (Table 2) were averaged to obtain $k_6$ (296 K) = (1.68 ± 0.12) × $10^{-10}$ $cm^3$ molecule$^{-1}$ s$^{-1}$.**

### 3.2    PLP-LIF determinations of $k_3(T)$

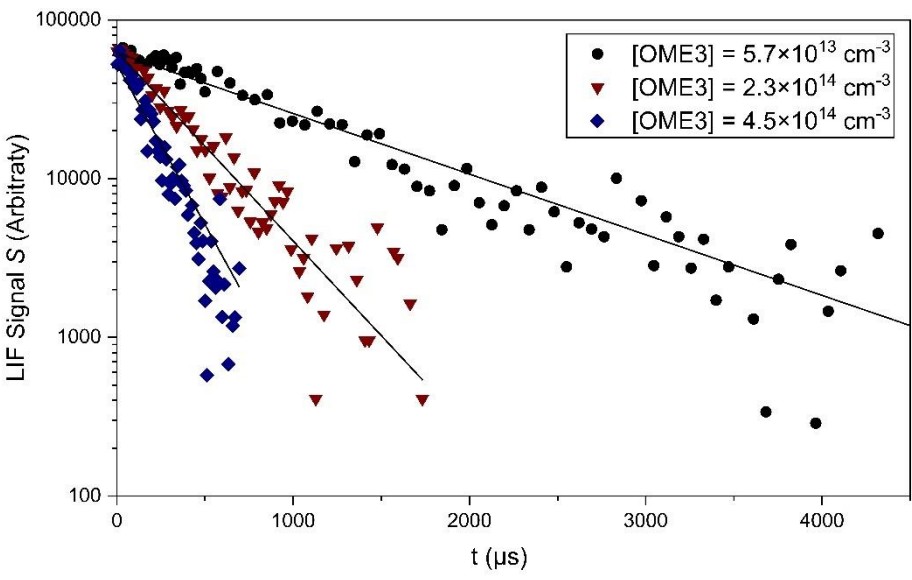





**Figure 4: Displays OH decays obtained in PLP-LIF experiments at $T$ = 297 K using three different [OME3]. Each was fit with Eq. (2) to determine pseudo first-order rate coefficients: $B$ = (964 ± 37) s$^{-1}$ at [OME3] = 5.7×10$^{13}$ molecule cm$^{-3}$; $B$ = (2755 ± 109) s$^{-1}$ at [OME3] = 2.3×10$^{14}$ molecule cm$^{-3}$; and $B$ = (5252 ± 220) s$^{-1}$ at [OME3] = 4.5×10$^{14}$ molecule cm$^{-3}$.**

PLP-LIF studies were carried under pseudo-first order conditions of [OME3] >> [OH] such that OH LIF time profiles, S(t), were described by a monoexponential decay expression (Eq. 2), following subtraction of measured baseline.

$$S(t) = S_0 e^{-Bt} \qquad (2)$$

$S_0$ describes the LIF signal at t = 0 in arbitrary units. It is proportional to the initial [OH] produced (R8) by the laser pulse. $B$ is the pseudo-first-order rate coefficient for OH decay, which includes components from both transport and reactive losses. Typical OH decay profiles are displayed in Figure 4, recorded in the presence of three different excess [OME3]. Other conditions ($P$ = 100 mbar (N$_2$), $T$ = 297 K, and [H$_2$O$_2$] ≈ 1×10$^{14}$ molecule cm$^{-3}$) were unchanged between experiments. OH LIF profiles were typically exponential over at least an order of magnitude and once fit to Eq. (2) yielded values of $B$ with high precision (standard errors were generally > 5%). Systematic errors from unintended radical side-reactions were considered unlikely. The low [OH] used ensured that OH losses by self-reaction or by reactions with the products of (R3) were minimal. Furthermore, aliphatic ethers do not have significant absorptions above 200 nm (Christianson et al., 2021), preventing the generation of organic radical fragments by the laser flash. Nevertheless, a series of experiments were performed in order to probe the impact of secondary chemistry. No systematic change in $B$ was obtained when the photolysis laser fluence was altered by a factor of three *via* modifications to the Q-switch delay. These observations suggest that any secondary chemistry had a negligible influence.

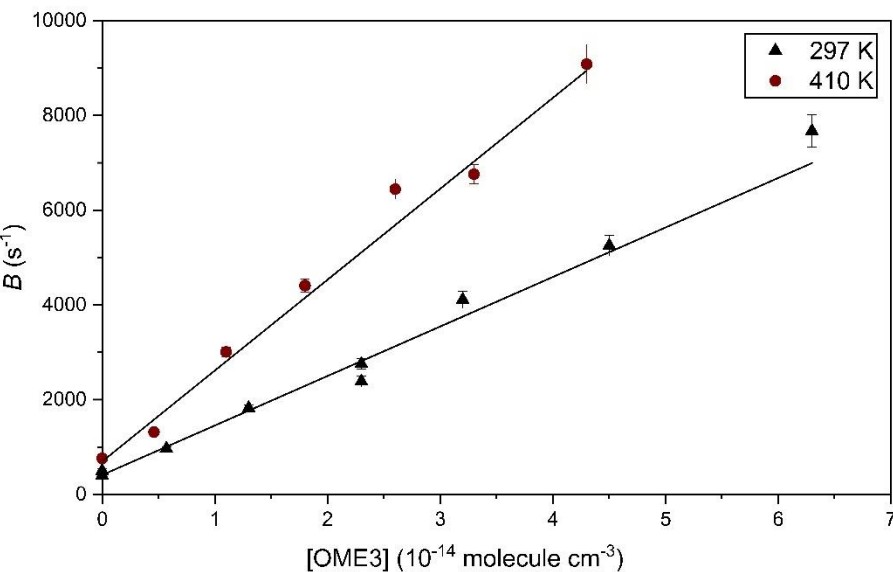

**Figure 5: Exemplary bimolecular plot used to determine $k_3(T)$ from absolute PLP experiments. The black triangles depict results at ambient temperature, used to obtain $k_3$(297 K) = (1.05 ± 0.05) × 10$^{-11}$ cm$^3$ molecule$^{-1}$ s$^{-1}$ *via* a weighted least squares linear fit of**





**Eq. 3 to the data. The red circles similar depict data obtained at elevated temperature to yield $k_3(410\ \text{K}) = (1.92 \pm 0.08) \times 10^{-11}\ \text{cm}^3$ molecule$^{-1}$ s$^{-1}$**

Figure 5 displays a plot of parameter $B$ vs. [OME-3] at $T = 297$ K; data were fit with Eq. (3) to obtain values of $k_3(T)$:

$$B = k_3[OME3] + k_{loss} \quad (3)$$

where the term $k_{loss}$ (in s$^{-1}$) corresponds to other losses of OH, in this case primarily the reaction of OH with the photolysis precursor $H_2O_2$ (R16) with small secondary contributions from diffusion and flow out of the reaction zone.

$$OH + H_2O_2 \rightarrow H_2O + HO_2 \qquad\qquad (R16)$$

When fitted linearly gradients correspond to $k_3(T)$ at each temperature and intercept values are in line with the predicted rate for (R16) with an estimated $[H_2O_2] = 10^{14}$ molecule cm$^{-3}$ (around a few hundred s$^{-1}$). A mean of four values obtained at $T = (298 \pm 2$ K) yielded $k_3 = (1.03 \pm 0.13) \times 10^{-11}$ cm$^3$ molecule$^{-1}$ s$^{-1}$ in both $N_2$ and air regardless of pressure. An analogous series of experiments, conducted at elevated temperatures, were used to determine $k_3(297 - 464$ K). Details of conditions employed and results from all such PLP-LIF experiments are listed in Table 3.

**Table 3: Absolute $k_3(T)$ determined *via* PLP-LIF in this work.**

| $T$ / K | $p$ / mbar | $n^a$ | [OME3]$^b$ | $k(T)^c$ |
|---|---|---|---|---|
| 297 | 78 | 16 | 1-37 | $1.05 \pm 0.05$ |
| 298 | 59 | 10 | 2-15 | $0.90 \pm 0.06$ |
| 300 | 56 | 8 | 2-18 | $0.96 \pm 0.06$ |
| 300 | 56 | 8 | 2-18 | $1.2 \pm 0.2$ |
| 337 | 76 | 8 | 2-25 | $1.00 \pm 0.06$ |
| 357 | 80 | 10 | 1-14 | $1.05 \pm 0.04$ |
| 373 | 74 | 8 | 1-12 | $1.21 \pm 0.12$ |
| 390 | 39 | 8 | 1-15 | $1.08 \pm 0.09$ |
| 410 | 33 | 13 | 1-14 | $1.92 \pm 0.08$ |
| 464 | 66 | 13 | 1-14 | $2.9 \pm 0.2$ |
| 469 | 33 | 13 | 1-14 | $2.3 \pm 0.2$ |
| Notes: a n = number of $B$ determinations, excluding those where [OME3] = 0; b = units of concentration were $10^{14}$ molecule cm$^{-3}$; c = units of k were $10^{-11}$ cm$^3$ molecule$^{-3}$ s$^{-3}$ | | | | |

Figure 6 represents all k3 results from this work in Arrhenius format. Whilst $k_3$(T) displayed approximately conventional Arrhenius behaviour at elevated temperatures, little change in the rate coefficient was observed between $T = 297$ K and $T = 390$ K. Potential explanations for this non-Arrhenius behaviour are explored in section 3.3 below, together with a comparison to results for similar oxygenated VOC + OH.





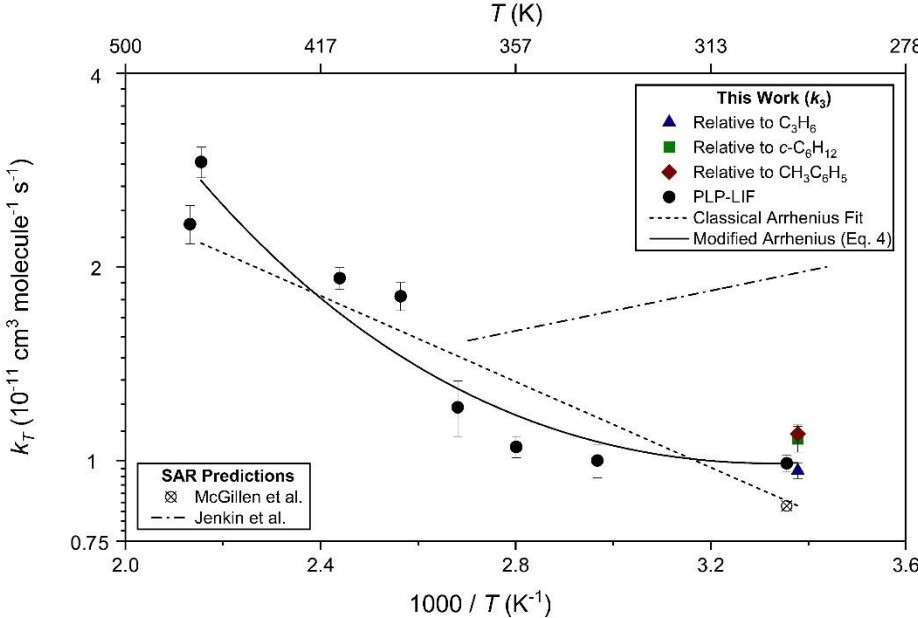

**Figure 6: Arrhenius plot displaying $k_3(T)$ results from this work obtained *via* PLP–LIF and relative rate experiments. Displayed as the dotted line is the poor fit to the classical Arrhenius expression and the solid line the fit to the modified Arrhenius expression; the dot-dash line depicts calculated $k_3$(290–370 K) using the SAR proposed by Jenkin et al. (2018). Error bars are representative of measurement reproducibility only. These data are further discussed in section 3.3.**

Experiments to determine accurate $k_4(T)$ in direct PLP-LIF experiments were not successful. Whilst reasonable exponential decays were recorded for the removal of OH in the presence of [OME4], nonlinearities were observed in the bimolecular plots of parameter $B$ vs. [OME4]. These observations were indicative of an inability to reliably saturate the gas flow through the bubbler with OME4, meaning that values of [OME4] calculated from manometric measurements were likely overestimates. Semi quantitative results were obtained in conditions when bubbler flows were smallest and the reaction cell was maintained at a slightly elevated temperature of $T$ = 340 K. Here the bubbler was most likely to saturate with OME4 to allow for accurate concentration determination, and condensation of OME4 in the reactor was avoided. This data indicated that (R4) proceeds with a rate coefficient $k_4(340\ \mathrm{K}) \approx 1 \times 10^{-11}\ \mathrm{cm}^3\ \mathrm{molecule}^{-1}\ \mathrm{s}^{-1}$, in line with the ambient temperature relative rate determinations from this work.

### 3.3 Discussion of Kinetic Results

Determinations of $k_3(298 \pm 2\ \mathrm{K}) = (1.03 \pm 0.13) \times 10^{-11}\ \mathrm{cm}^3\ \mathrm{molecule}^{-1}\ \mathrm{s}^{-1}$ (PLP-LIF) and $k_3(296 \pm 2\ \mathrm{K}) = (1.04 \pm 0.09) \times 10^{-11}\ \mathrm{cm}^3\ \mathrm{molecule}^{-1}\ \mathrm{s}^{-1}$ (relative rate), between the different methods agree well with one another, especially when considering systematic uncertainties. These kinetic methods rely upon different critical assumptions (absolute knowledge of [OME-3] for



PLP-LIF, reference rate coefficients for relative-rate) and have complementary strengths and weaknesses, lending a high degree of confidence to these results. Nevertheless, when considering the systematic uncertainties inherent to these experiments (exact [OME] for the PLP-LIF determinations; the significant uncertainties in reference rate coefficient values for the relative rate experiments) we consider quoting a more conservative overall $k_3(297 \pm 3$ K$) = (1.0 \pm 0.2) \times 10^{-11}$ cm$^3$ molecule$^{-1}$ s$^{-1}$ appropriate. Rate coefficients for (R4) were determined solely from relative rate experiments, giving an averaged value of $k_4(296 \pm 2$ K$) = (1.11 \pm 0.10) \times 10^{-11}$ cm$^3$ molecule$^{-1}$ s$^{-1}$. While complementary PLP-LIF experiments were not possible due to the limited volatility of OME4, semi-quantitative results indicate a rate coefficient $k_4(340$ K$) \approx 1 \times 10^{-11}$ cm$^3$ molecule$^{-1}$ s$^{-1}$, in support of the relative rate results. Given the increased uncertainty surrounding this value we recommend a value of $k_4(297 \pm 3$ K$) = (1.1 \pm 0.4) \times 10^{-11}$ cm$^3$ molecule$^{-1}$ s$^{-1}$. To the best of our knowledge, these results represent the first reported kinetic data for (R3-R4) under atmospherically relevant conditions.

In the absence of prior experimental studies of (R3- R4), it is difficult to meaningfully critique these values further. Table 4 presents a comparison of ambient temperature k-values for the reactions of OME and related compounds with OH. There is a clear trend whereby the larger compounds react with OH faster than those with fewer CH$_2$ groups. This trend is in line with the basic principle of SAR, whereby each additional oxidisable group contributes to additional reactivity. However, calculations based on an established SAR (Jenkin et al., 2018) appear to overpredict the reactivity of these molecules, for example $k_{3Jenkin}(298$ K$) = 1.9 \times 10^{-11}$ cm$^3$ molecule$^{-1}$ s$^{-1}$. This overprediction from the Jenkin SAR appears common to (R3) and (R4) and the available literature for OH + OME1 (Table 4). A similar overprediction was recently observed for OH + 2,2,5,5-tetramethyloxolane (TMO) (Mapelli et al., 2022). A lack of suitable literature data has likely limited progress on training SAR for OH + ether reactions. Rate coefficients for these reactions can also be estimated by an extension of the electrotopological approach which has proved an effective for estimating the reactivity of OH with alkanes and haloalkanes (McGillen et al., 2024). This method works by considering both the electronic state and connectivity of each atom in a molecule. This approach can be adapted to ethers by including a consideration for hydrogen-bonded pre-reaction complexes, a common feature of OH + oxygenate reactions, giving the values listed in Table 4 (Max McGillen, personal communication). This method proves more effective than the SAR from Jenkin et al. in terms of both absolute value and observed trends across the series.

**Table 4: A comparison of predicted and experimental kinetic data for OH + OME and related compounds.**

| OH + | k / 10$^{-11}$ cm$^3$ molecule$^{-1}$ s$^{-1}$ | | | | Notes on experimental values |
|---|---|---|---|---|---|
| | **Jenkin** | **McGillen** | **COSMO-RS** | **Experiment** | |
| **CH$_3$OCH$_3$** | 0.26 | - | 0.42 | 0.28 | Recommended value from IUPAC evaluation with $\Delta \log k = 0.08$ (or ~ 20%)(Atkinson et al., 2006). The absolute rate coefficient studies that form the basis of the preferred values (Arif et al., 1997; Bonard et al., 2002; Mellouki et al., 1995; Tully |





| | | | | | and Droege, 1987), are in good agreement with the other two absolute temperature-dependent studies (Perry et al., 1977; Wallington et al., 1988). |
|---|---|---|---|---|---|
| **OME1** | 0.82 | - | 0.40 | (0.44 ± 0.02) | (Bänsch and Olzmann, 2019) |
| | | | | (0.46 ± 0.16) | (Wallington et al., 1997), absolute |
| | | | | (0.55 ± 0.06) | (Wallington et al., 1997), relative to $c$-C$_6$H$_{12}$ |
| | | | | (0.50 ± 0.04) | (Wallington et al., 1997), relative to ethylene |
| | | | | (0.49 ± 0.01) | (Porter et al., 1997), absolute |
| | | | | (0.49 ± 0.02) | (Thuner et al., 1999), relative to $c$-C$_6$H$_{12}$ |
| **OME2** | 1.38 | - | 0.42 | - | Not reported to the best of our knowledge |
| **OME3** | 1.94 | 0.85 | 0.44 | (1.0 ± 0.2) | This work, (R3), PLP-LIF and RR |
| **OME4** | 2.50 | 0.99 | 0.49 | (1.1 ± 0.4) | This work, (R4), RR only |

COSMO-RS was used to estimate values (Table 4) based on a combined DFT-parameterisation approach (Klamt, 1993, 1996, 2018). This method was able to estimate rate constants to a reasonable accuracy without the requirement for training on related structures, however it struggles to predict trends along the series from CH$_3$OCH$_3$. This could be due to extremely similar charge density surfaces across the series or due to deviation from the pre-reaction complex formation assumed for all oxygenates by the model. The proprietary nature of the software makes more thorough interpretation of these results challenging. Evidence for the likely participation of hydrogen-bonded complexes was provided by the results obtained with PLP-LIF for $k_3$ over the range of temperatures 294K to 464 K (Figure 6). Conventional Arrhenius-like behaviour was observed for $k_3(\geq400$ K). The region of unchanging $k_3(297 – 390$ K) may be indicative of an increasingly important role for hydrogen-bonded pre-reaction complexes at moderate temperatures. Experimental constraints meant we could not confirm these observations *via* the relative rate method, except at ambient temperature. Such "U-shaped" Arrhenius plots have been observed for OH reactions with oxygenated VOC, notably dimethyl ether (Arif et al., 1997; Bonard et al., 2002; Mellouki et al., 1995), acetone (Wollenhaupt et al., 2000), methyl pivalate (Mapelli et al., 2023; Wallington et al., 2001), diethyl ether, a series of di-alkoxy methanes (Calvert et al., 2020) and most recently TMO (Mapelli et al., 2022). We were able to achieve a good fit to the modified Arrhenius expression Eq. (4) used by Rutto et all to describe OH + cyclopentanone and derivatives (Rutto et al., 2024):

$$k(T) = A \left(\frac{T}{300}\right)^{\alpha} e^{\left(-\frac{E}{R}\left(\frac{1}{T} - \frac{1}{300}\right)\right)} \quad (4)$$

Where $A = (9.9 \pm 0.3) \times 10^{-12}$ cm$^3$, $\alpha = (12.4 \pm 1.5)$ (dimensionless) and E/R = (-3717 ± 534) K. We note that, by analogy with the other reactions of OH with oxygenates noted in section 3.3 above, and as implied by Eq. 4, $k_3$ may increase at lower temperatures and that this $k_3(297$ K) value used here may be an underestimate of $k_3$ in cooler atmospheric conditions, e.g. the upper troposphere. It was not possible to investigate this more fully, given the lack of suitable low-temperature apparatus in York or Iasi. Also displayed on Fig. 4 are $k_3(294 – 370$ K) calculated using the Jenkin et al. SAR, which clearly over-predicts



experimental $k_3$ values and exhibits more distinctive non-Arrhenius behaviour, and $k_3$(298 K) calculated using the SAR from McGillen.

Results from the OME + Cl determinations were $k_{6,RR}$(296 K) = $(1.68 \pm 0.12) \times 10^{-10}$ cm$^3$ molecule$^{-1}$ s$^{-1}$ and $k_{7,RR}$(296 K) = $(1.85 \pm 0.18) \times 10^{-10}$ cm$^3$ molecule$^{-1}$ s$^{-1}$. Given inherent uncertainties in reference rate coefficients used here, the lack of complementary PLP-LIF experimental data, or indeed any previous literature on we suggest conservative values of $k_6$(296 K) = $(1.7 \pm 0.4) \times 10^{-10}$ cm$^3$ molecule$^{-1}$ s$^{-1}$ and $k_7$(296 K) = $(1.9 \pm 0.6) \times 10^{-10}$ cm$^3$ molecule$^{-1}$ s$^{-1}$ to more fully account for uncertainties. These rate coefficients are approximately an order of magnitude larger than the corresponding OME + OH values. Such enhanced reactivity towards Cl is not unusual for many classes of VOC, including ethers; reactions of Cl with THF (Alwe et al., 2013; Szymański and Sarzyński, 2020) and with dioxane (Giri et al., 2011; Li and Pirasteh, 2006) are similarly rapid. Again, to the best of our knowledge, this work presents the first such determinations of $k_6$ and $k_7$.

## 4    Atmospheric Implications and Conclusions

Atmospheric lifetimes ($\tau$) for OME3 and OME4 were estimated using Eq. 5, using [OH] = $1.13 \times 10^6$ molecule cm$^{-3}$ representative of the troposphere (Lelieveld et al., 2016) in combination with rate coefficients from this work. The resultant $\tau_3 \approx 24$ hours and $\tau_4 \approx 22$ hours may be overestimates for two reasons. Firstly our ambient-temperature value may underestimate $k_3$ for colder tropospheric conditions. Secondly, the lifetime of OME may be constrained by (R6-7) with chlorine atoms in regions highly impacted by atmospheric chlorine. Estimates for tropospheric [Cl] are subject to a high degree of spatial variability and vary widely, however using a large estimate of [Cl] = $1 \times 10^4$ molecule cm$^{-3}$ (Li et al., 2018) and OME + Cl rate coefficients from this work, slightly shorter overall (OH and Cl reaction) lifetimes were estimated: $\tau_6 \approx$ 21 hours for OME3; $\tau_7 \approx$ 20 hours for OME4. Losses of OME to other atmospheric radicals, to O$_3$, and to photolysis are , to date, unreported and appear unlikely and as such we conclude that reactions (R3-4) are the dominant process for OME removal from the troposphere. Moriarty et al. (2003) used Eq. 5 to estimate lifetimes, $\tau_1 \approx 25$ hours and $\tau_2 \approx 16$ hours, for the breakdown of dioxane and THF respectively. OME3 and OME4 have similar lifetimes to the solvents they may replace and so to better quantify their potential air quality impacts, Photochemical Ozone Creation Potential estimates ($POCP_E$) were calculated by the method reported by Jenkin et al. (2017). This parameter measures the extent of ozone generation, relative to an equivalent mass of ethene (CH$_2$CH$_2$), of a given VOOC. $POCP_E$ were calculated from the structure of each compound, and the rate coefficient for reaction with OH. $POCP_E$ values in of NW-European atmospheric conditions were estimated to be 26 (OME3) and 23 (OME4) by this method. Analogous estimates reveal significantly larger $POCP_E$ for dioxane and THF (40 and 55 respectively). These results suggest that OMEs would have a smaller negative impact on air quality than the traditional solvents they could replace. The rationale for this discrepancy is that, whilst OME have a similar reactivity to other traditional solvents, they are already more oxidised, with no C-C and fewer C-H bonds (per kg) available to fuel atmospheric oxidation cycles.



To conclude, the application of two distinctive experimental approaches has revealed that OME3 and OME4 react with OH (R3 - R4) slower than predicted by the state of the art SAR. A SAR currently-under-development from McGillen has proven more accurate when confronted with this structurally distinct class of compounds. These alternative solvents have similar atmospheric lifetimes yet appreciably smaller $POCP_E$ than the traditional, problematic solvents they could replace. These findings lend further strength to the growing case for OMEs as greener alternatives to THF and dioxane. Further, results here represent a significant expansion to the extremely limited kinetic database for reactions of polyoxygenated VOC with OH. In the course of this work rate coefficients for both OMEs with atomic chlorine were determined, which are similar to THF and dioxane. These and similar results may allow for the training of improved SAR and similar predictive tools in future.

## 5    Data Availability

Data pertaining to Gaussian and COSMO-RS calculations (S1), exemplar PLP-LIF data (S2) OME separation and characterisation (S3) and relative rate studies (S4), are included in the supplement. Any other data are available upon request from the corresponding authors.

## 6    Author Contributions

Laser-based experiments were designed by TJD and conducted by TJD and CM. Chamber experiments were designed by TJD, IGB, CA and RIO and conducted by JDM, RKW and CR. Gaussian and COSMO-RS calculations were carried out by RKW and JDM respectively. The manuscript was written by TJD and JDM with assistance from other authors. TJD, IGB and CRM conceived of the overall project.

## 7    Competing Interests

The authors declare that they have no conflict of interest.

## 8    Ackowledgements

The chamber experiments from this work received funding from a Transnational access project that is supported by the European Commission under the Horizon 2020 – Research and Innovation Framework Programme, H2020-INFRAIA-2020-1, ATMO-ACCESS Grant Agreement number: 101008004. JDM would like to thank Merck KGaA and the Dept. of Chemistry at the University of York for his jointly funded PhD Studentship, Royal Society of Chemistry for Researcher Travel and Development Grant No. D23-4336052173 to enable his lab visit to UAIC, Dr James Sherwood for support given during his PhD project and Dr Suranjana Bose for her technical and logistical assistance. CM similarly thanks the Dept. of Chemistry at York for a PhD scholarship. RKW would like to thank the Natural Environment Research Council (NERC),





grant number NE/S007458/1 for funding her PhD Studentship. CR, CA, RIO and IGB acknowledge support from PN-III-P2-2.1-PED-2021-4119 and PN-III-P4-PCE2021-0673 UEFISCDI projects.

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
