# Peer review of "Atmospheric breakdown kinetics and air quality impact of potential "green" solvents the oxymethylene ethers OME3 and OME4"

_EGUsphere, 2025_

## Referee Comment (RC2)

**Review of "Atmospheric breakdown kinetics and air quality impact of potential "green" solvents the oxymethylene ethers OME3 and OME4"**

Author(s): James D. D'Souza Metcalf et al.

This study is relevant to the field of atmospheric chemistry, as it provides the first experimental rate coefficients for reactions of interest. These data are important for the development of improved structure–activity relationships or other predictive tools for kinetic parameters describing OH + VOC reactions.

I recommend publication after addressing the comments below.

The text lacks clarity at times, and the language should be improved in several places. The experiments and theoretical work require further elaboration, particularly to demonstrate that appropriate validation tests have been performed. Additionally, the discussion of the results needs to be more thoroughly developed.

The manuscript also lacks a single, well-structured document titled *Supplementary Information*, which should include clearly defined subsections detailing the computational methods, as well as the experimental and theoretical results. Currently, the supporting information is spread across multiple files that are not described and some of them are not easily accessible. A dedicated section at the end of the main manuscript should summarize the contents of the Supplementary Information.

**Specific comments**

- line 62: It should be added that the R1 − R4 reactions are hydrogen abstraction reactions.

- line 71: The year for Moriarty et al is missing.

- line 77: Please add the acronym used in reaction R5 in the text, i.e. "2,2,5,5-tetramethyloxolane (TMO)". Suggest mentioning that this compound is a substituted tetrahydrofuran to explain why it has been chosen as an example and adding the values for the rate coefficients predicted by SAR and found experimentally.

- line 80: replace "lab-based" with "laboratory-based"

- line 83: "further indication into the atmospheric fate of…" Please replace "indication" with "insight".

- line 98: Please add the name of the compound labelled with OME5.

- line 100 (Figure 1): Some of the trace colours are very similar, making it difficult to identify which compound they correspond to. Please choose more distinct colours to improve clarity.

Please add the OME4 spectrum to the figure.

- line 102 (capture of Figure 1): "(OME3)" is missing after 989 cm-1, this wavenumber is not attributed to any compound.

- line 105-106: I think it is necessary that the authors comment about the possible losses of OME3 and especially OME4 in liquid form on the walls of septum/chamber during injection as those compounds have low vapour pressure. Were any tests performed to verify whether the gas phase concentrations of the OMEs matched those calculated based on the volume of liquid injected into the chamber? Such tests could involve injecting different volumes of the OME3/OME4 and checking whether the resulting optical absorbance ratios correspond to the volume ratios.

- line 107: Please replace C2O2Cl2 with (COCl)2

- line 113 (reaction R9): A small fraction of COCl is formed. Please comment if there are any issues about sequential decomposition of COCl.

- line 117: There is a partial overlap between the absorption features of OME3 and C3H6 (figure 1). OME3 was measured at 989 cm-1 where there is some absorption of C3H6. Can the author clarify how this was taken into account when determining the 989 cm-1 absorbance of OME3?

The errors in the determination of concentrations caused by spectral overlap can be avoided by fitting reference spectra to the measured spectra in a range of wavenumbers. Could the authors comment on their choice of determining concentrations from $A_{max}$ instead of using the fitting method?

- line 126: Citations are missing after "from literature". Why OME spectra predicted through DFT calculations were used instead of experimental spectra, determined by the authors?

- line 132 (table 1): It is unclear what is meant by "Fitted or manually entered data from multiple sources" in the Source of recommended values column.

-line 146: What is the estimated concentration of $H_2O_2$ that led to the stated approximate concentration of OH?

- lines 158-171: Although line 93 mentions that kinetic parameters were determined via computational methods, Section 2.4 does not provide an explanation of the procedure.

- line 185 (Figure 2 caption): As it is the first time when the notation $k_{3RR}$ is used it should be mentioned that it represents the rate coefficient obtained by the relative rate method. Please replace "exemplar plot" with "example of a plot" or "typical/representative plot", which are more widely used alternatives; the same for figures 3 and 5.

- line 205 (Figure 4): Please provide a plot with the entire decays, showing their baselines at later times. I suggest showing the current Figure 4 as an inset of this figure. Fitting the entire experimental decays will show whether they follow a single exponential behaviour as assumed.

- line 206: I suggest delete the word "Displays" and change "each was …" with "each decay was"

- line 207: The typical notation is "$k_{obs}$", not "$B$" as the meaning is the observed rate coefficient.

- lines 217-218: Suggest adding "(see below)" after "Systematic errors from unintended radical side-reactions were considered unlikely."

- line 247: Please add "(equation 4)" after "modified Arrhenius expression".

- lines 262-263: The absolute knowledge of [OME] is required by the rate relative method too, not only by the PLP-LIF method.

- line 265: This line states that there are significant uncertainties in reference rate coefficient values. However, according to table 1 the errors are typically 10-20%.

- line 271: An error of about 40% is recommended for k4. Can the authors elaborate on the methodology used for this estimation? The same for the conservative errors of the other rate coefficients.

- line 275: The authors states that "There is a clear trend whereby the larger compounds react with OH faster than those with fewer CH2 groups". However, table 4 shows that while the OH + OME1 reaction is nearly twice as fast as the OH + $CH_3OCH_3$ reaction, OME3 and OME4 react with OH at similar rates. Can the authors comment on this?

- line 288: The values for CH3OCH3, OME1 and OME2 determined by McGillen's method are needed in order to conclude if this method gives overall better predictions than the SAR of Jenkin et al. (2018). Table 4 shows that the Jenkin et al.'s prediction of k(OH+CH3OCH3) is close to the experimental value. The authors should comment on that. The comparison given in table 4 would benefit if calculations using different SARs would be included.

- line 289: Can the authors explain the principle of the COSMO-RS method? The text states that "the pre-reaction complex formation assumed for all oxygenates by the model". Please explain what pre-reaction complex formation was assumed as a number of publications are cited (Klamt, 1993, 1996, 2018).

- line 309: "Also" at the beginning of the sentence should be replaced by "In addition" or "Additionally"

- lines 314-315: Please see the comment about line 271 above.

- line 320: The sections Atmospheric Implications and Conclusions need to be separated

- line 321: Equation 5 is not given

- line 334: I think VOOC is a typo and it should be VOC

- lines 339-340: Can the authors specify which traditional solvents are being referred to? Please re-phrase "fuel atmospheric oxidation cycles."

---

## Author Comment (AC1)

**Author Response to Reviewer Comments**

Reviewer Comments in *Italic Blue*. Author responses in **Bold Black.**

**Opening Comments**

**We would like to take this opportunity to thank both reviewers for their constructive critique and insightful comments. We believe our revised manuscript is significantly stronger than our initial submission as a result of their input.**

**Reviewer 1**

Opening comments:

*In general, the experimental study presents in this work, using two techniques at different temperatures for the OH case, seems very interesting to me. I believe this study should be published, as it provides valuable kinetic data and represents a significant expansion to the extremely limited kinetic database for reactions of polyoxygenated VOCs with OH.*

**We are grateful to the reviewer for their positive feedback.**

*However, the discussion of the results appears to be quite weak and should have been explored in more depth.*

**We value the reviewers' suggestions. The discussion has now been updated and expanded.**

*In some cases, the supplementary information cannot be accessed without the appropriate software. The computational calculation section lacks detail.*

**We agree that the submitted form of the supplementary information could be significantly improved and have now addressed this. Due to some of the software involved it has not always been possible to provide files in a universally accessible format however we have wherever possible made improvements.**

*I do not understand why it is necessary to perform computational calculations to determine the IR spectra of the studied compounds (OM3 and OM4) and why MOPAC is used for the calculation of the rate constant instead of ab initio calculations.*

**We have provided more detailed discussion of our computational methods here and updated the main text as appropriate.**

Minor comments:

*Line 21: I do not think it is necessary to identify the reaction as R3 in the abstract.*

*Line 89: I suggest using capital initials or clearly identifying the acronyms.*

*Line 96: Please indicate the acronyms for ESI-MS*

*Figure 1: X-axis legend: The unit cm $^{-1}$ is typically identified as wavenumber, not frequency (s $^{-1}$)*

*Line 101: Figure 1 caption: Please indicate the characteristic IR band for OME3.*

*Line 245: Figure 6. Review the PLP-LIF data. I believe the data presented in the graph at 390K does not correspond to the data in Table 3. Why were all values close to room temperature not represented?*

*Line 305: Review the units of A.*

*Line 320: Where is equation 5?*

*Line 330: Again, equation 5 is referenced, which does not appear in the text.*

*Line 334: Define VOOC.*

**We concur with the above suggestions, all of which have been addressed in our updated manuscript.**

*Lines 68-69: Is Moriarty et al. 2003 considered the most recent reference?*

**Moriarty et al., 2003 is the most recent experimental rate constant determination, and also provides a self-consistent set of lifetime estimates for each OVOC of interest. The more recent references, such as the book from Calvert et al., (2015) all cite this value.**

*Line 95: Why was the extraction of these compounds carried out instead of working with commercial compounds?*

**Members of our group have previously worked on the solvent application of these ethers, focusing both on blends, (as these are readily commercially available in large quantities), but also characterising the individual compounds (Zhenova et al., 2019). As a result of this, we have access to both large quantities of the blend and robust procedures for their separation. It was therefore more practical for us to carry out distillations than try to source the individual compounds from specialty suppliers. All OME samples we used in our experiments were of high (>97%) purity by GC-FID.**

*Line 98: Could you clarify why it is referred to as S3 rather than S1, given that this is the first time it is cited in the document?*

**This was done in error and has been fixed as part of the wider reworking of the supplementary information suggested by Reviewer 2.**

*Line 126: To what extent are the bands of the spectrum obtained through computational calculations reliable? Why were experimental spectra from real samples not used?*

**Experimental spectra were recorded for each OME and each reference compound (as displayed in Figure 1), and used for final selection of reference compounds. QCC estimated spectra were used to 'pre-screen' potential reference compounds before we began experiments to ensure that valuable time was not wasted recording spectra for wholly inappropriate compounds.**

*Lines 168-169: I do not understand how the theoretical calculation of the rate coefficient (k) is performed. It would be helpful to include an example of the calculation in the supplementary material. Typically, the theoretical rate coefficient is calculated using the Transition State Theory (TST).*

**We absolutely agree more detail on the method of calculation would be valuable, however the COSMOTherm software is proprietary and effectively functions as a black box – we put optimised structures in, it gives us back rate constants. It offers no options to adjust the parameters used in the calculation and only accepts structures optimised using the related, and similarly opaque, COSMOConf software. The exact details of how the software works have not been made public (to clarify - the MOPAC calculations are contained within the COSMOThermX software, at no point does the user interact directly with a MOPAC interface) beyond it citing the two 1990's papers we referenced in the main document (Klamt, 1993, 1996), it is unclear if the methodology has been updated since then. From the above cited papers we can be sure that it is not using Transition State Theory (TST), instead calculating parameters form the properties of the molecular orbitals calculated by COSMOConf, and feeding these parameters into a series of empirically derived relationships to get to the rate constant. This approach was useful for estimating OH + VOC rate constants, particularly for multifunctional oxygenates which lie outside the training sets of most SARs. We have added a table and figure to the reworked SI document comparing its outputs to some IUPAC recommended values to demonstrate this. One key advantage of this software is that it is commonplace among chemists who work on green alternative solvents. It therefore provides an accessible starting point for such researchers to consider**

the atmospheric impact of their molecules early in their design processes (often before they have even been synthesised) which is something we are actively trying to encourage.

*Line 278: Why not use the AOPWIN program, based on Atkinson's SAR? The calculated value for R3 predicts a k of 1.19x10-11 cm3 molecule-1 s-1, very similar to the experimental one.*

This is a very valid critique, however we would, respectfully, prefer to stick to our decision to prioritise the Jenkin SAR. The Jenkin SAR is a significant update and rework of the decades-old Atkinson SAR that has, in our experience largely replaced it in many applications. The fact that the Atkinson SAR gets closer on this occasion is essentially a chance occurrence, not one resulting from a superior model. Indeed its original authors question its validity specifically ethers and polyethers and say *"The present estimation technique is reasonably reliable when used within the database used in its derivation, but extrapolation to organic compounds outside of this database results in a lack of assurance of its reliability, and its use for organic compounds which belong to classes other than those used in its development is discouraged."*(Kwok and Atkinson, 1995). In order to maintain the concision of our discussion we opted to compare only to the most up-to-date iteration of the SAR, the recent and methodologically distinct electrotopological state approach from McGillen and the predictive tool most available to green chemists, COSMO-RS. Bringing up what is essentially an outdated version of the Jenkin SAR only to then question its relevance and validity would, in our view, add little to the discussion other than to confuse it.

*Line 288: Table 4. Why not include the formula of all compounds (OME1, OME2; OME3, OME4), as in the case of CH3OCH3? Why not give values for this compound according (OME1 and OME2) to McGillen's method?*

Table 4 has now been updated to include the full set of predicted values from the McGIllen SAR. We agree the notation in table 4 could be improved and have thus relabelled the first compound as "*OME0 (dimethyl ether)*" to ensure uniformity and legibility.

*Lines 296-296: Regarding to the paragraph "The region of unchanging k3(297 – 390 K) may be indicative of an increasingly important role for hydrogen-bonded pre-reaction complexes at moderate temperatures" Could you provide more detail about these complexes?*

**We have updated and reordered the text to read:**

"*The region of unchanging k₃(297 – 390 K) may be indicative of a change in mechanism across this temperature range. Hydrogen bonded pre-reaction complexes have previously been suggested to explain similar observations in the case of other OH + oxygenated VOC reactions. Such "U-shaped" Arrhenius plots have been observed for OH reactions with oxygenated VOC, notably dimethyl ether (Arif et al., 1997; Bonard et al., 2002; Mellouki et al., 1995), acetone (Wollenhaupt et al., 2000), methyl pivalate (Mapelli et al., 2023; Wallington et al., 2001), diethyl ether, a series of di-alkoxy methanes (Calvert et al., 2020) and most recently TMO (Mapelli et al., 2022).*"

*Line 307: I do not understand this statement: "that this k3(297 K) value used here may be an underestimate of k3 in cooler atmospheric conditions."*

**We believe that this curve is unlikely to remain flat at temperatures below room temperature. Results for many other OH + OVOC reactions would indicate that the k(T) curve could pass through a minimum close to ambient temperature with values likely to increase as the temperature is lowered. There is strong literature precedent for this such as those listed in our reply to the previous comment.**

*Lines 316-319: I believe the manuscript would benefit from a more comprehensive discussion of the results in comparison to the values reported in the literature for similar compounds.*

**We have now expanded this section with discussion of additional compounds, although the available literature on polyethers specifically is quite limited.**

*Lines 323-324: Regarding the paragraph indicated in these lines, I believe the problem is not that k3 is underestimated. I consider that the k3 value calculated at 297 K is correct; the problem is that this value has been used as if it were the one corresponding to the temperature in the troposphere. I suggest rephrasing the sentence to make the idea clearer.*

**This sentence has now been rephrased as "*Firstly, our ambient-temperature value may underestimate k₃ for conditions throughout the troposphere where temperatures below 297 K are common*"**

*Line 342: If Atkinson's SAR is used, the constant value for R3 is very similar to the experimental one.*

Atkinson's SAR gives a good prediction for OME3, but not OME4, unlike the McGillen SAR which predicts both well. As previously explained, we do not feel the Atkinson SAR is the most relevant for comparison. Respectfully, we would prefer to keep this section as it is.

**Reviewer 2**

Opening comments:

*This study is relevant to the field of atmospheric chemistry, as it provides the first experimental rate coefficients for reactions of interest. These data are important for the development of improved structure–activity relationships or other predictive tools for kinetic parameters describing OH + VOC reactions. I recommend publication after addressing the comments below.*

**We are grateful to the reviewer for their feedback and recommendation.**

*The text lacks clarity at times, and the language should be improved in several places. The experiments and theoretical work require further elaboration, particularly to demonstrate that appropriate validation tests have been performed. Additionally, the discussion of the results needs to be more thoroughly developed.*

**We are appreciative of these suggestions, which we have addressed individually below.**

*The manuscript also lacks a single, well-structured document titled Supplementary Information, which should include clearly defined subsections detailing the computational methods, as well as the experimental and theoretical results. Currently, the supporting information is spread across multiple files that are not described and some of them are not easily accessible. A dedicated section at the end of the main manuscript should summarize the contents of the Supplementary Information.*

**We thank the reviewer for this valuable suggestion, and have submitted an updated and reordered SI alongside the new manuscript.**

Minor comments:

*line 71: The year for Moriarty et al is missing.*

*line 77: Please add the acronym used in reaction R5 in the text, i.e. "2,2,5,5-tetramethyloxolane (TMO)". Suggest mentioning that this compound is a substituted tetrahydrofuran to explain why it has been chosen as an example and adding the values for the rate coefficients predicted by SAR and found experimentally.*

*line 80: replace "lab-based" with "laboratory-based"*

*line 83: "further indication into the atmospheric fate of…" Please replace "indication" with "insight".*

*line 98: Please add the name of the compound labelled with OME5.*

*line 102 (capture of Figure 1): "(OME3)" is missing after 989 cm-1, this wavenumber is not attributed to any compound.*

*line 107: Please replace C2O2Cl2 with $(COCl)_2$*

*line 185 (Figure 2 caption): As it is the first time when the notation k3RR is used it should be mentioned that it represents the rate coefficient obtained by the relative rate method. Please replace "exemplar plot" with "example of a plot" or "typical/representative plot", which are more widely used alternatives; the same for figures 3 and 5.*

*line 205 (Figure 4): Please provide a plot with the entire decays, showing their baselines at later times. I suggest showing the current Figure 4 as an inset of this figure. Fitting the entire experimental decays will show whether they follow a single exponential behaviour as assumed.*

*line 206: I suggest delete the word "Displays" and change "each was ..." with "each decay was"*

*line 207: The typical notation is "kobs", not "B" as the meaning is the observed rate coefficient.*

*line 247: Please add "(equation 4)" after "modified Arrhenius expression"*

*line 321: Equation 5 is not given*

*line 334: I think VOOC is a typo and it should be VOC*

*lines 339-340: Can the authors specify which traditional solvents are being referred to? Please re-phrase "fuel atmospheric oxidation cycles."*

**We concur with the above suggestions, all of which have been addressed in our updated manuscript.**

*line 62: It should be added that the R1 – R4 reactions are hydrogen abstraction reactions.*

**We have added the phrase "all of which likely proceed *via* H abstraction to an organic radical fragment"**

*line 100 (Figure 1): Some of the trace colours are very similar, making it difficult to identify which compound they correspond to. Please choose more distinct colours to improve clarity.*

*Please add the OME4 spectrum to the figure.*

We agree that the legibility of figure 1 could be improved. We have opted to update the entire figure to a 'stacked' format, which we believe significantly improves readability. OME4 has now been added.

*line 105-106: I think it is necessary that the authors comment about the possible losses of OME3 and especially OME4 in liquid form on the walls of septum/chamber during injection as those compounds have low vapour pressure. Were any tests performed to verify whether the gas phase concentrations of the OMEs matched those calculated based on the volume of liquid injected into the chamber? Such tests could involve injecting different volumes of the OME3/OME4 and checking whether the resulting optical absorbance ratios correspond to the volume ratios.*

The relative rate has the advantage of not requiring knowledge of OME and reference compound concentrations. The correction rate constants for the wall loss, photolysis and any other dilutions may be required to accurately calculate rate coefficients, however tests for possible losses from adsorption on surfaces and later diffusion from the walls or inlet ports have been performed and were considered for correction. Additional text has been included to explain possible artefacts and the way the tests were performed.

*line 113 (reaction R9): A small fraction of COCl is formed. Please comment if there are any issues about sequential decomposition of COCl.*

We do not believe sequential decomposition presents a significant issue for the relative rate experiments. The small fraction of COCl that is formed by the photolysis of (COCl)$_2$ rapidly decomposes to CO + Cl at room temperature rather than persisting long enough to interfere with the intended VOC + Cl chemistry as it might if used in a flash or pulsed laser photolysis experiment.

*line 117: There is a partial overlap between the absorption features of OME3 and C3H6 (figure 1). OME3 was measured at 989 cm-1 where there is some absorption of C3H6. Can the author clarify how this was taken into account when determining the 989 cm-1 absorbance of OME3?*

*The errors in the determination of concentrations caused by spectral overlap can be avoided by fitting reference spectra to the measured spectra in a range of wavenumbers. Could the authors comment on their choice of determining concentrations from Amax instead of using the fitting method?*

There is a partial overlap of some absorption features of propene and OME3 at 989 cm$^{-1}$. However, OME3 was also monitored also by additional feature at 1128 cm$^{-1}$ and both were considered when determining scaling factors. The good linearity of the relative rate plots (as in figures 2 and 3) for all reference compounds employed in this study and the agreement of the rate coefficients obtained between the three reference compounds shows that the scaling factors obtained through the subtraction procedure were not affected by peak overlap.

*line 126: Citations are missing after "from literature". Why OME spectra predicted through DFT calculations were used instead of experimental spectra, determined by the authors?*

The phrase "from literature" as used in error, in fact we compared to reference spectra taken in the ESC-Q-UAIC chamber. We have updated the text to reflect this.

Please see our response to reviewer one discussing the use of calculated IR spectra.

*line 132 (table 1): It is unclear what is meant by "Fitted or manually entered data from multiple sources" in the Source of recommended values column.*

This is how the data are described in the EUROCHAMP database, which we have taken to mean that the compiler of the database collated and fitted a range of existing literature data to arrive at their recommended value. There is little guidance available on the EUROCHAMP site and as to the exact meaning of this phrase and thus we have opted to quote it without any (potentially spurious) interpretation. We have placed this phase in quotation marks to reflect this.

*line 146: What is the estimated concentration of $H_2O_2$ that led to the stated approximate concentration of OH?*

We estimate [$H_2O_2$] of around $2\times10^{14}$ molecules cm$^{-3}$ lead to our suggested OH concentration. Using this concentration with our estimated 20 mJ cm$^{-2}$ gives [OH] = $5\times10^{11}$ molecules cm$^{-3}$. This [$H_2O_2$] also fits with the concentration inferred by the intercept of figure 5. We have updated the text with a suggested [$H_2O_2$] of $\lesssim 2\times10^{14}$ molecules cm$^{-3}$.

*lines 158-171: Although line 93 mentions that kinetic parameters were determined via computational methods, Section 2.4 does not provide an explanation of the procedure.*

We direct the reviewer to our response to reviewer 1 on the same topic.

We would also like to state that the procedure provided should be sufficient for an unfamiliar user to reproduce our calculations exactly, as the software involved is straightforward and we have outlined all required steps.

*lines 217-218: Suggest adding "(see below)" after "Systematic errors from unintended radical side-reactions were considered unlikely."*

We have opted to add the phrase "*given the following observations and checks*" given that said checks are detailed immediately after the above quoted section in the text.

*lines 262-263: The absolute knowledge of [OME] is required by the rate relative method too, not only by the PLP-LIF method.*

The advantage of the relative rate method is that no absolute concentrations of the OMEs and reference compounds is required. Any parameters directly proportional to concentration can be used to determine the logarithmic representations of OME:Reference depletion ratio.

*line 265: This line states that there are significant uncertainties in reference rate coefficient values. However, according to table 1 the errors are typically 10-20%.*

We agree that this choice of language is a little misleading and have therefore removed the word "significant".

*line 271: An error of about 40% is recommended for k4. Can the authors elaborate on the methodology used for this estimation? The same for the conservative errors of the other rate coefficients.*

The 20% error in $k_3$ stems from the largest uncertainties present in each measurement technique; we suggest an error in [OME] of around 15% in the PLP-LIF experiments and have tabulated errors of around 15-25% in the reference reactions used for the relative determinations. As such, whilst our statistical errors for each measurement are in the region of 10%, we believe a 20% error better accounts for the aforementioned uncertainties. This was doubled in the case of $k_4$ as it could only be measured by one technique.

*line 275:* *The authors states that "There is a clear trend whereby the larger compounds react with OH faster than those with fewer CH2 groups". However, table 4 shows that while the OH + OME1 reaction is nearly twice as fast as the OH + CH$_3$OCH$_3$ reaction, OME3 and OME4 react with OH at similar rates. Can the authors comment on this?*

While OME3 and OME4 react at roughly the same rate, they both in turn react roughly twice as fast as OME1 which in turn reacts about twice as fast as dimethyl ether. This is the basis of our comment that *"There is a clear trend whereby the larger compounds react with OH faster than those with fewer CH2 groups"*. We have added the phrase "*although this effect is lessened as the compounds become larger such that the difference between OME3 and OME4 is small-to-negligible*"

*line 288:* *The values for CH3OCH3, OME1 and OME2 determined by McGillen's method are needed in order to conclude if this method gives overall better predictions than the SAR of Jenkin et al. (2018). Table 4 shows that the Jenkin et al.'s prediction of k(OH+CH3OCH3) is close to the experimental value. The authors should comment on that. The comparison given in table 4 would benefit if calculations using different SARs would be included.*

We agree with the reviewer that the McGillen SAR predictions for the full series of compounds would be valuable here, and they have now been added. It is unsurprising to us that dimethyl ether is predicted accurately by the Jenkin SAR as it is included in the training set (Jenkin et al., 2018). We have now updated the text to reflect this. Whilst we agree that an argument can be made in favour of including more SARs and predictive techniques, in our view there is a balance to be struck between completeness and concision. We direct reviewer 2 to our response to reviewer 1's comments on the same topic for further discussion.

*line 289:* *Can the authors explain the principle of the COSMO-RS method? The text states that "the pre-reaction complex formation assumed for all oxygenates by the model". Please explain what pre-reaction complex formation was assumed as a number of publications are cited (Klamt, 1993, 1996, 2018).*

We would first direct reviewer 2 to our response to reviewer 1's comments on the same topic.

We would like to add the following to address reviewer 2's question regarding COSMO-RS' treatment of pre-reaction complexes:

 Klamt (1996) describes in extensive detail how the hydrogen bonded complexes are accounted for. In brief, an additional parameter is added to the molecular orbital activity relationship, the components of which include descriptors of the

distance between the abstractable hydrogen and any nearby lone-pairs and a factor describing the propensity of each class of oxygen-containing functionalities to form hydrogen bonds. It is stated in the same paper than ether groups are particularly difficult to account for and that under some circumstances they are better accounted for by the earlier iteration of the model (Klamt 1993) that does not account for hydrogen bonded complexes at all. Given the large number of ethereal oxygen atoms in OME3 and OME4 it appears likely to us that this difficulty in accounting for ethereal hydrogen bonding may explain the decrease in accuracy from DME to OME4. Without detailed information on what updates have been made to the model in in the intervening years, particularly since its inclusion in the CosmoThermX package, it is difficult to assemble a meaningful discussion of these issues without significant speculation, hence our statement "*The proprietary nature of the software makes more thorough interpretation of these results challenging*".

*line 309: "Also" at the beginning of the sentence should be replaced by "In addition" or "Additionally"*

We respectfully disagree and believe in this case "A*lso*" is appropriate and flows better than "*In addition*" or "*Additionally*". We would thus prefer to leave the text as-is.

*lines 314-315: Please see the comment about line 271 above.*

Please see our reply to the same comment.

*line 320: The sections Atmospheric Implications and Conclusions need to be separated*

With respect, we are happy with the current sections and layout. The atmospheric implications and conclusions are heavily interlinked. If separated they would, in our view, form unnecessarily short sections with a worsened narrative structure.

**References**

Arif, M., Dellinger, B., and Taylor, P. H.: Rate Coefficients of Hydroxyl Radical Reaction with Dimethyl Ether and Methyl *tert* -Butyl Ether over an Extended Temperature Range, J. Phys. Chem. A, 101, 2436–2441, https://doi.org/10.1021/jp963119w, 1997.

Bonard, A., Daële, V., Delfau, J.-L., and Vovelle, C.: Kinetics of OH Radical Reactions with Methane in the Temperature Range 295−660 K and with Dimethyl Ether and Methyl-

*tert* -butyl Ether in the Temperature Range 295–618 K, J. Phys. Chem. A, 106, 4384–4389, https://doi.org/10.1021/jp012425t, 2002.

Calvert, J. G., Orlando, J. J., Stockwell, W. R., and Wallington, T. J.: The Mechanisms of Reactions Influencing Atmospheric Ozone, Oxford University Press, https://doi.org/10.1093/oso/9780190233020.001.0001, 2015.

Calvert, J. G., Mellouki, A., Orlando, J. J., Pilling, M. J., and Wallington, T. J.: The mechanisms of atmospheric oxidation of the oxygenates, Oxford University Press, New York, 1 pp., https://doi.org/10.1093/oso/9780199767076.001.0001, 2020.

Jenkin, M. E., Valorso, R., Aumont, B., Rickard, A. R., and Wallington, T. J.: Estimation of rate coefficients and branching ratios for gas-phase reactions of OH with aliphatic organic compounds for use in automated mechanism construction, Atmospheric Chem. Phys., 18, 9297–9328, https://doi.org/10.5194/acp-18-9297-2018, 2018.

Kwok, E. and Atkinson, R.: Estimation of hydroxyl radical reaction rate constants for gas-phase organic compounds using a structure-reactivity relationship: An update, Atmos. Environ., 29, 1685–1695, https://doi.org/10.1016/1352-2310(95)00069-B, 1995.

Mapelli, C., Schleicher, J. V., Hawtin, A., Rankine, C. D., Whiting, F. C., Byrne, F., McElroy, C. R., Roman, C., Arsene, C., Olariu, R. I., Bejan, I. G., and Dillon, T. J.: Atmospheric breakdown chemistry of the new "green" solvent 2,2,5,5-tetramethyloxolane via gas-phase reactions with OH and Cl radicals, Atmospheric Chem. Phys., 22, 14589–14602, https://doi.org/10.5194/acp-22-14589-2022, 2022.

Mapelli, C., Donnelly, J. K., Hogan, Ú. E., Rickard, A. R., Robinson, A. T., Byrne, F., McElroy, C. R., Curchod, B. F. E., Hollas, D., and Dillon, T. J.: Atmospheric oxidation of new "green" solvents – Part 2: methyl pivalate and pinacolone, Atmospheric Chem. Phys., 23, 7767–7779, https://doi.org/10.5194/acp-23-7767-2023, 2023.

Mellouki, A., Teton, S., and Le Bras, G.: Kinetics of OH radical reactions with a series of ethers, Int. J. Chem. Kinet., 27, 791–805, https://doi.org/10.1002/kin.550270806, 1995.

Moriarty, J., Sidebottom, H., Wenger, J., Mellouki, A., and Le Bras, G.: Kinetic Studies on the Reactions of Hydroxyl Radicals with Cyclic Ethers and Aliphatic Diethers, J. Phys. Chem. A, 107, 1499–1505, https://doi.org/10.1021/jp021267i, 2003.

Wallington, T. J., Ninomiya, Y., Mashino, M., Kawasaki, M., Orkin, V. L., Huie, R. E., Kurylo, M. J., Carter, W. P. L., Luo, D., and Malkina, I. L.: Atmospheric Oxidation Mechanism of Methyl Pivalate, $(CH_3)_3CC(O)OCH_3$, J. Phys. Chem. A, 105, 7225–7235, https://doi.org/10.1021/jp010308s, 2001.

Wollenhaupt, M., Carl, S. A., Horowitz, A., and Crowley, J. N.: Rate Coefficients for Reaction of OH with Acetone between 202 and 395 K, J. Phys. Chem. A, 104, 2695–2705, https://doi.org/10.1021/jp993738f, 2000.

Zhenova, A., Pellis, A., Milescu, R. A., McElroy, C. R., White, R. J., and Clark, J. H.: Solvent Applications of Short-Chain Oxymethylene Dimethyl Ether Oligomers, ACS

Sustain. Chem. Eng., 7, 14834–14840,
https://doi.org/10.1021/acssuschemeng.9b02895, 2019.